# Bone Regeneration Assessment of Polycaprolactone Membrane on Critical-Size Defects in Rat Calvaria

**DOI:** 10.3390/membranes11020124

**Published:** 2021-02-09

**Authors:** Ana Paula Farnezi Bassi, Vinícius Ferreira Bizelli, Tamires Mello Francatti, Ana Carulina Rezende de Moares Ferreira, Járede Carvalho Pereira, Hesham Mohammed Al-Sharani, Flavia de Almeida Lucas, Leonardo Perez Faverani

**Affiliations:** 1Department of Diagnosis and Surgery, School of Dentistry, São Paulo State University—UNESP, Araçatuba, São Paulo 16015-050, Brazil; v.bizelli@unesp.com (V.F.B.); mf_tamires@hotmail.com (T.M.F.); caru_moraes@hotmail.com (A.C.R.d.M.F.); institutojarede@gmail.com (J.C.P.); leonardo.faverani@unesp.br (L.P.F.); 2School of Dentistry, Faculty of Dentistry, Ibb University, Ibb, Yemen; hishamm2010@live.com; 3Department of Maxillofacial Surgery, School of Stomatology, Harbin Medical University, Harbin 150081, China; 4Araçatuba Veterinary Medicine School, UNESP—São Paulo State University, Araçatuba, São Paulo 16050-680, Brazil; flavia.lucas@unesp.br

**Keywords:** membranes, polymers, bone regeneration, materials biocompatibility, materials testing, polycaprolactone, bone substitute

## Abstract

Biomaterials for use in guided bone regeneration (GBR) are constantly being investigated and developed to improve clinical outcomes. The present study aimed to comparatively evaluate the biological performance of different membranes during the bone healing process of 8 mm critical defects in rat calvaria in order to assess their influence on the quality of the newly formed bone. Seventy-two adult male rats were divided into three experimental groups (n = 24) based on the membranes used: the CG—membrane-free control group (only blood clot, negative control), BG—porcine collagen membrane group (Bio-Guide^®^, positive control), and the PCL—polycaprolactone (enriched with 5% hydroxyapatite) membrane group (experimental group). Histological and histometric analyses were performed at 7, 15, 30, and 60 days postoperatively. The quantitative data were analyzed by two-way ANOVA and Tukey’s test (*p* < 0.05). At 7 and 15 days, the inflammatory responses in the BG and PCL groups were significantly different (*p* < 0.05). The PCL group, at 15 days, showed a large area of newly formed bone. At 30 and 60 days postoperatively, the PCL and BG groups exhibited similar bone healing, including some specimens showing complete closure of the critical defect (*p* = 0.799). Thus, the PCL membrane was biocompatible, and has the potential to help with GBR procedures.

## 1. Introduction

Current implant systems are increasingly using methods for reconstruction of the alveolar process to allow optimal positioning of implants for esthetic and functional purposes [1,2,3,4,5,6].

Guided bone regeneration (GBR) [7] is commonly used to treat conditions that reduce alveolar bone defects in the jaw [8,9,10]. 

GBR is characterized by the use of physical media (membranes or barriers) that promote the anatomical isolation of a site, allowing the proliferation of a group of cells—predominantly osteoblasts—from the receptor site and simultaneously preventing the action of inhibitory factors on the process of regeneration [11].

Currently, resorbable collagen membranes dominate GBR procedures in clinical practice. They have several advantages over non-absorbable membranes as they stabilize the wound, allow early vascularization, attract fibroblasts through chemotaxis, and are semipermeable, which facilitates the transfer of nutrients. The hydrophilic properties of the membranes facilitate the surgical technique and stabilization of the graft [12,13,14,15,16].

The membranes can also support cell growth because their structure mimics the extracellular matrix, thus allowing specific growth, proliferation, migration, and tissue differentiation for future implantation [17]. The extracellular matrix is composed of an interconnection of protein fibers (collagen and elastin), proteoglycans, and mineral deposits in the case of bone tissue [18]. However, this technique is still in development, so studies have to be conducted to test these biomaterials used in GBR. In a rat animal model, critical defects [19,20] at the calvaria allow for the study of the real biological behavior of these biomaterials.

Two of the polymers receiving the most research attention at present are polylactic acid (PLA) and polycaprolactone (PCL). Both are biodegradable and biocompatible [21,22]. Recent studies have shown that membranes based on a PCL matrix associated with different substances can provide efficient support for cellular development, allowing for the regeneration of cartilage, vascular tissue, cardiac tissue, skin tissue, eye tissue, and nerve tissue, among others [23,24,25,26]. However, PCL is more hydrophobic than PLA [27,28,29,30] and exhibits slower degradation in saline solutions [31]. One study demonstrated that when hydroxyapatite (HA) was added, the molecular weight of PCL increased dramatically, while the molecular weight of PLA dropped [30]. Higher molecular weight leads to a slower degradation time of the polymer, enabling safer guided tissue regeneration. The mechanical properties of these membranes may influence cell growth, and their morphology may influence cell adhesion. It is possible to incorporate proteins into polymers [32,33], improving the properties of PCL.

Biodegradable polymer membranes with nanohydroxyapatite (nHA) [34,35,36,37] have been studied in recent years because their structure can be maintained for longer periods and is more favorable for osteoblasts. The nHA constitutes almost 70% of the bone tissue and can be used in association with polymers to promote rapid bone neoformation in the GBR procedures by inducing stem cells to differentiate into osteoprogenitor cells. The advantage is that the rate of degradation is controllable, maintaining the structure for a longer period of cell growth, and the sheet has nanometric voids between the nanofibers that can anchor the cells. The nanofibers can be processed to be porous, inducing the crystallization of the nHA [35,36] by suitable treatment, or they can be surface-treated with plasma [38] to increase their wettability, which increases the anchorage and the degree of compatibility of matrix cells. To improve the dispersion of nHA, compatibilizers might be used [39], as they show a better response in osteoblasts. In some studies [40], it was found that the concentration of nHA can strengthen or create defects in the nanofibers; low concentrations showed better results [41].

As the mechanical properties of the membranes can influence cell adhesion and growth, mechanical improvements at the scaffold matrix are advantageous. Pure PCL nanofibers demonstrated worse results in mechanical tests than incorporated PCL nanofibers with some concentrations of nHA [17,42,43].

The objective of this study was to evaluate the potential of the PCL membrane with 5% HA in the bone healing of critical bone defects in rat calvaria.

## 2. Materials and Methods

### 2.1. Development of the Polycaprolactone Membrane 

A PCL sheet associated with the nHA material was developed by the Department of Materials Engineering, UFSCar, São Carlos, SP, Brazil. For the preparation of the PCL solution with the nanocharge (0.05 g/mL), the nHA was first left in chloroform and ultrasonically cleaned (Ultrasonic Cleaner, model USC 1450, São Paulo, SP, Brazil) for 90 min to promote better dispersion of the nanoparticles. Then, the polymer was added. After the polymer completely dissolved, methanol was added under constant stirring until complete homogenization of the solution. The solvent system was composed of 75% chloroform and 25% methanol (*v*/*v*). The concentration of the polymer in this suspension, consisting of the polymer and nanocharger, was 0.12 g/mL. The polymer used was Solvay’s PCL CAPA 6806 with an 80 kDa molar mass [44,45].

### 2.2. Samples

This research was approved by the Ethical Committee in the Use of Animals from Araçatuba Dental School, UNESP under protocol number # 2013–0148, and followed the ARRIVE Guidelines (Animal Research: Reporting of in Vivo Experiments) [46].

Seventy-two adult male (3 to 4 months) rats (*Rattus novergicus albinus*, Wistar), weighing approximately 200–300 g, were kept in cages containing four animals per cage. They were kept under controlled conditions, with a dark/light cycle, and fed with a balanced ration (NUVILAB, Curitiba PR, Brazil—1.4% Ca and 0.8% P) and water ad libitum. The rats were randomly divided in to three groups (n = 24 per group): CG—membrane-free control group (only blood clot, negative control), BG—porcine collagen membrane group (Bio-Guide^®^, positive control), and PCL—polycaprolactone (enriched with 5% hydroxyapatite) membrane group (experimental group) in the Vivarium of the Araçatuba Dental School, UNESP. On the day of the surgery, in each animal, a critical bone defect (8 mm) was made in the calvaria. The website www.lee.dante.br (accessed on 29 December 2020) was used to calculate the sample size; for the calculation, the standard deviation used was 12.5, the difference of means was 80%, and *p* was <0.05, with four animals per group per period [47,48].

After the surgical procedure, the animals were euthanized at four time points during the experiment, at 7, 15, 30, and 60 days postoperatively, as described below:Clot group (CG) (negative control): The critical bone defect was filled with blood clot.Porcine collagen group (Bio-Guide^®^) (BG) (positive control): The critical bone defect was filled with blood clot and a porcine collagen membrane was placed on the defect.PCL group with 5% HA (PCL) (experimental group): The critical bone defect was filled with blood clot and a PCL membrane with 5% HA was placed on the defect.

### 2.3. Experimental Surgical Procedure

The animals were made to fast for 12 h preoperatively and were subjected to sedation through the intraperitoneal administration of ketamine hydrochloride (Francotar-Virbac do Brasil Ltda, São Paulo, Brazil) with xylazine (Rompun, Bayer AS- Animal Health, São Paulo, Brazil) at a dosage of 50 mg/kg and 5 mg/kg, respectively [49,50,51,52,53].

An aseptic protocol was adopted, which included the sterilization of the instruments and surgical fields used to delimit the area of operation, and the use of sterile surgical gowns and gloves. All surgical procedures were performed in the operation room of the Vivarium of the Araçatuba Dental School, UNESP. Trichotomy was conducted in the calvaria region, antisepsis with topical PVPI (PVPI 10% Riodeine, Rioquímica, São José do Rio Preto).

After performing a V-shaped occipitofrontal incision of 2 cm and a total detachment of the flap, a critical surgical defect of 8 mm diameter [54] was made in the central portion of the calvaria, maintaining the dura mater’s integrity. For that, a 7 mm diameter trephine drill bit (3i Implant Innovations, Inc., Palm Beach Gardens, FL, USA) was used, coupled with a low speed of 500 rpm under heavy irrigation with 0.9% sodium chloride solution (Darrow, Rio de Janeiro, Brazil). According to the proposed treatments, the defects were filled with blood clot, with the addition of a porcine collagen membrane (BG), or a PCL membrane with 5% HA (PCL with 5% HA), or none (CG), placed on the defect.

At the end of the procedure, the soft tissues were sutured in planes. At immediately the post-operatively, each animal received a single intraperitoneal dose of 0.2 mg/mL penicillin G benzathine (Pentabiotico Veterinário Porte, Fort Dodge Saúde Animal Ltda., Campinas, SP, Brazil). Every two days, the animals were carried out with clean cages and food [49,50,51,52,53].

### 2.4. Histological and Histometric Analysis

All samples collected after euthanasia of the animals were fixed in 4% paraformaldehyde for 48 h for tissue fixation. After that, the samples were washed in water and decalcified in 10% ethylenediaminetetraacetic acid (EDTA) for 5 weeks. The samples were washed for 12 h and dehydrated in different concentrations of alcohol (70% and 100%), followed by diaphanization in xylol, and included in paraffin. Finally, the samples were cut in a microtome with 5 µm between each slice. The histological blades were mounted and stained with hematoxylin and eosin [55,56,57,58,59,60,61,62].

After laboratory processing, the histological sections to be analyzed were selected and photomicrographs of the blades were made under an image processing system, which consisted of a light microscope (DM 4000 B, Leica, Mannheim, Germany), a color image processor (Leica Qwin V3, Leica software, Mannheim, Germany), a color camera (DFC 500, Leica, Mannheim, Germany), and a computer (Intel Core I5, Intel Corp, Santa Clara, CA, USA; Windows 10, Microsoft Corp, Redmond, WA, USA), with the ImageJ digitized image analyzer software (National Institutes of Health, Bethesda, MD, USA). Histometric analysis was performed through inflammatory cell and vessel count; the inflammatory response was determined, and the neoformed bone area was measured.

During the blades analysis, the examiner was blinded and had their identifications hidden. For the inflammatory cell and vessel count, two sections were analyzed for each animal (total 48) and three regions were evaluated: the center of the defect, the right side, and the left side of the defect. In the original objective of ×100, 130 points were predetermined, and the ones that touched a cell were counted. Two sections per animal were used to evaluate and measure the total area (TA) of the bone defect and the newly formed bone area (NFB) in μm^2^ in the center of the defect (primary outcome). TA was considered 100% of the area and NFB a percentage of TA. The data obtained in the analyses were transformed from absolute values of pixels to percentage values to allow for the application of the statistical test [49,50].

### 2.5. Statistical Analysis

The data collected from the histometric analysis were tested using SigmaPlot 12.0 software (Exact Graphs and Data Analysis, San Jose, CA, USA), with a 5% level of significance. Initially, the Shapiro–Wilk normality test was applied, which noted homogeneity of the data as a function of the normality curve (*p* > 0.05). Thereafter, the two-way ANOVA (experimental groups and analysis periods) was applied. For both interactions, a statistically significant change (*p* < 0.05) was observed, in which the Tukey post-test was applied. For data from immunohistochemical analyses, the scores were subjected to variance analysis, two-way ANOVA test, and Holm–Sidak post-test, in consideration of the sources of variation (membranes and periods of analysis).

## 3. Results

No complications were observed during the experimental surgical procedure or postoperatively, therefore, none of the animals were excluded from this study.

### 3.1. Membranes’ Osteopromotive Histological Behavior 

The cytoarchitecture of the microscopic sections of the initial period (7 days) demonstrated an organized inflammatory infiltrate and NFB tissue, with intense osteoblastic activity close to the stumps for all groups. In that same period, there was also an intense fibroblastic activity in the CG group. At 15 days, all groups demonstrated improvement in osteopromotive properties. In the BG and PCL groups, an organized connective tissue and some osteoid sites at the center of the defect and near the stumps were observed.

At 30 days, only the BG group presented neoformed bone tissue in the central portion of the defect; however, the PCL group demonstrated an excellent performance, reducing the cavity size and presenting a large area of neoformed bone. Areas of organized connective tissue were observed in the areas where bone neoformation was not present. In the CG group, the extent of the defect was almost entirely filled with loose connective tissue. In the final period of repair (60 days), the CG group continued with the expected pattern, with bone neoformation only in the peripheral portion of the defect. The PCL test group demonstrated, in some specimens, the complete closure of the defect, and in others, a nearly complete closure. As expected, the BG group showed results very similar to the 30-day experimental period, but with a much larger area of NFB tissue (Figure 1).

### 3.2. Inflammatory Response: Lymphocyte and Vessel Counting

Regarding the PCL membrane’s behavior during the analyzed periods, a statistically significant difference was observed in the BG group (*p* < 0.05) between 7 and 15 days in inflammatory cells, with a focus on lymphocytes and vessels. The BG and PCL groups’ biological behaviors were similar, demonstrating a decrease in lymphocytes between 7 and 15 days postoperatively, and an increase in vessels, but the intensity of the response in the PCL group was lower. At 7 days, a statistically significant difference was observed in the number of vessels (*p* = 0.037), but at 15 days, the PCL group demonstrated good biological behavior for angiogenesis, and no statistically significant difference was observed (*p* = 0.144). No statistically significant difference was observed for the inflammatory cells at 7 and 15 days (*p* = 0.052 and *p* = 0.214), respectively (Figure 2 and Figure 3).

### 3.3. Histometry: Newly Formed Bone Area

From the histological data obtained for the area of neoformed bone (NFB), at 7 days of bone repair, there was a similarity between the CG and PCL groups (*p* = 0.874), and the BG group demonstrated a better initial response with statistically significant difference in the bone healing process (*p* = 0.005). However, at 15 days postoperatively, the PCL group showed a statistically significant improvement compared to the BG group (*p* = 0.012), representing a large area of neoformed bone. In the final periods, 30 and 60 days, the PCL and BG groups had similar results with no statistically significant difference (*p* = 0.532 and *p* = 0.443, respectively). However, it was observed that the PCL group had a distinctly better result in the final process of bone repair (PCL > BG > CG) (Figure 4).

## 4. Discussion

At 7 days, all groups presented very little bone neoformation and only close to the stump. The connective tissue formed was very disorganized. The remainder of the membranes in the PCL and BG were well sharp and extensive, occupying all the centers of the defect.

In the 30-day period, a less favorable result was found in the CG, where the animals received only the clot, presenting small bone formation in the stumps and no bone neoformation in the center of the defect. On the other hand, the samples in the PCL and BG groups presented bone neoformation in the stumps and in the central region of the defects, NFB was observed as “islands“ fused with bone tissue and remnants of the membrane [63]. The samples of the PCL group also presented remnants of the membrane and the best results, since in addition to the new bone formation from the stumps, the NFB was more continuous.

In the 60-day period, the CG continued to present the worst results [49], with no bone neoformation at the center of the defect, only connective tissue (CT) defect filling. BG group animals presented better bone formation at 60 days, with almost complete closure of the defect, with only a small band in the center with CT and remnants of the membrane. The PCL group presented continuous bone neoformation throughout the defect, which was more uniform than that in the BG group. The CT was already more organized, and the membrane was absorbed by the organism during this period. In some specimens, there was a complete closure of the bone defect—an event that was not verified in the other groups [49,50].

The results of the CG group prove that the critical-size defect [54] does not allow spontaneous healing; therefore, the osteopromotive factor of the membranes was tested.

PCL and collagen membranes are nontoxic to cells and have several desirable properties, such as biodegradability [64]. This characteristic of low toxicity was proven in the present study since there were no histological characteristics in the evaluated periods that could indicate rejection of the inserted material and a low-intensity inflammatory process, with an intense reduction of the inflammatory cells from 7 to 15 days postoperatively. They also displayed an excellent result in the osteogenic factor, which is important for providing oxygen and nutrients for the osteoprogenitor cells [49,50].

In the present study, the PCL + HA membrane presented good histological results, due to the addition of hydroxyapatite and its osteogenic and osteopromotive factors [41,65]. In other studies, HA added to PCL nanofibers significantly improved bone healing in induced defects in the calvaria. Chen and Chang [66], for example, using nanofibers very similar to those used in the present study (PCL + HA and PCL), observed that they did not negatively affect osteogenic differentiation, allowing the growth and differentiation of mesenchymal cells. In addition, they observed that mineral production by the cells was proportional to the concentration of HA in the nanofiber. Conversely, Ródenas-Rochina et al. [67] observed that HA did not improve cell proliferation and osteoblast differentiation when added to PCL-based substrates, and PCL alone presented better results.

PCL has some drawbacks; it is hydrophobic, which leads to low affinity and cell adhesion [68]. In addition, it can produce acid byproducts during degradation, causing an inflammatory response [69]. However, this adverse event was not observed during the study period.

Ceramic and porous HA demonstrated the ability to integrate into the receptor bed, being osteoconductive and successful in the reconstruction of bone failures in medical and dental areas. A composite such as HA added to the membrane would improve the biocompatibility and integration of the hard tissue because the ceramic particles incorporated into the polymer matrix allow for a rapid increase in the initial propagation of serum with proteins, compared to the more hydrophobic polymer surface [70].

Synthetic and natural polymers undergo different degradation pathways. Synthetic polymers are generally degraded by simple hydrolysis, whereas natural polymers are mainly enzymatically degraded [71].

Local stresses are needed to stimulate bone neoformation. Conversely, excess stiffness of the membranes can cause stress concentrations in the surrounding healthy tissue, resulting in bone resorption [71].

Studies involving in vivo methodologies with PCL are rare, but in vitro studies with only the PCL prepared with electrospinning have demonstrated favorable results, corroborating the present study [72]. Therefore, research on biomaterials has received increasing investment over the years. There is a high expectation for the improvement of these grafts as the search to minimize the incidences of bone defects continues.

## 5. Conclusions

The PCL membrane is a biocompatible material that aided the GBR of critical defects in the calvaria of rats. However, the porcine collagen membrane presented a larger area of NFB tissue.

## Figures and Tables

**Figure 1 membranes-11-00124-f001:**
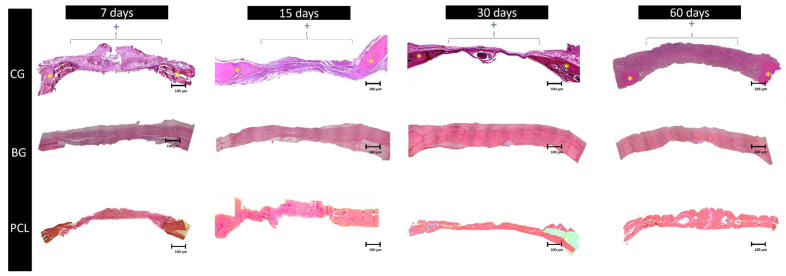
Photomicrographs at the panoramic reconstruction of histological sections (×6.3) for the experimental groups (CG (clot group), BG(Bio-Gide^®^ group), and PCL(polycaprolactone group)) in all periods analyzed (7, 15, 30, and 60 days postoperatively). The BG and PCL groups performed better at 60 days, demonstrating the ability to close the bone defect. The PCL group showed good closure of the defect, with particles of polycaprolactone and nanohydroxyapatite (nHA) surrounded by osteoid tissue. Asterisks (*) represent regions of bone stumps; plus signs (+) represent the horizontal extension of the bone defect.

**Figure 2 membranes-11-00124-f002:**
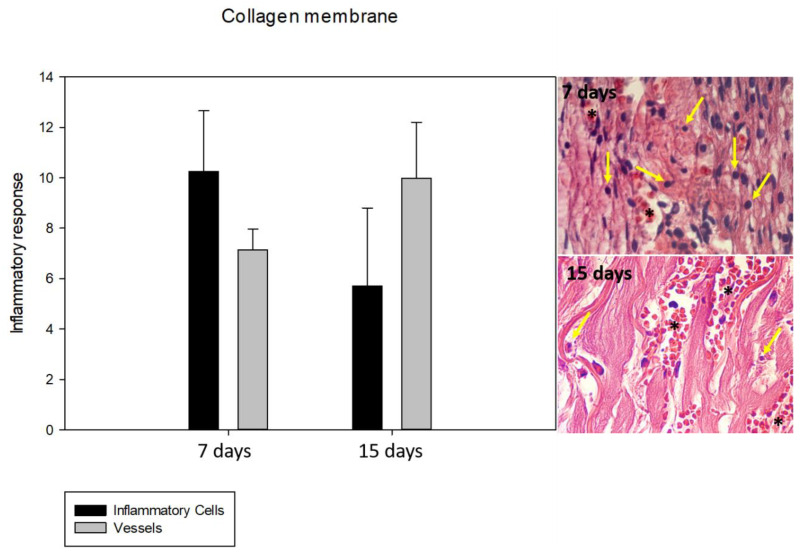
Graph and photomicrographs (×100) demonstrating the average number of inflammatory cells and vessels in the BG group during the periods analyzed. Regarding the biological behavior of the collagen membrane, there was a decrease in the number of inflammatory cells and an increase in the number of vessels. * indicates blood vessel.

**Figure 3 membranes-11-00124-f003:**
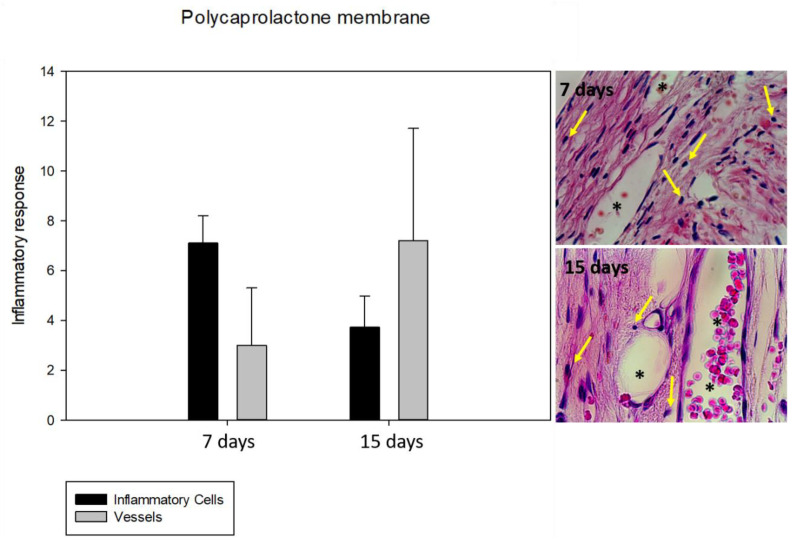
Graph and photomicrographs (×100) demonstrating the average number of inflammatory cells and vessels in the BG group during the periods analyzed. Regarding the biological behavior of the polycaprolactone membrane, there was a decrease in the number of inflammatory cells and an increase in the number of vessels. * indicates blood vessel.

**Figure 4 membranes-11-00124-f004:**
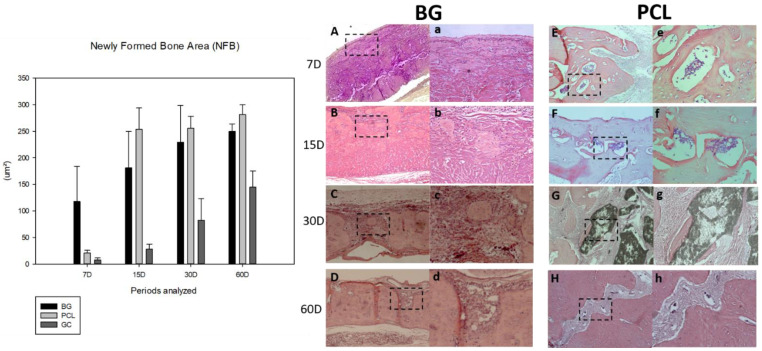
Graph and representative photomicrographs comparing the areas of bone neoformation between the groups and periods analyzed. Comparing the membranes used, there was no statistically significant difference between the PCL and BG groups (*p* = 0.799). The factor time, mainly at the initial periods, is critical for better results. For the BG group at 7 (**A**), 15 (**B**), 30 (**C**), and 60 (**D**) days (×12.5): (**A**) Panoramic image of newly formed bone near the periosteum. (**B**) Panoramic image of osteoid tissue islands at the center of the defect. (**C**) Panoramic image of the center of the defect almost closed with immature bone tissue. (**D**) Panoramic image of the center of the defect. (**a**–**d**) are higher magnifications (×25) of the panoramic views. For the PCL group (×12.5): (**E**) Panoramic image of the osteoid tissue around a polycaprolactone particle near the stump. (**F**) Panoramic image of the osteoid tissue around a polycaprolactone particle near the center of the defect. (**G**) Panoramic image of a large polycaprolactone particle with mature osteoid tissue. (**H**) Panoramic image of the center of the defect with a remaining particle of polycaprolactone. (**e**–**h**) are higher magnifications (×25) of the panoramic views. In short, PCL = BG > CG.

## Data Availability

Data available on request.

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
