# Peer review of "Bone Regeneration Assessment of Polycaprolactone Membrane on Critical-Size Defects in Rat Calvaria"

_membranes, 2021, doi:10.3390/membranes11020124_

Round 1

Reviewer 1 Report

The paper describes experimental data for bone regeneration assessment of PCL membrane on defect of rats calvaria. The histological results are not clear, and the control group was not well chosen. And the actual data present are not enough for a publication. I hope the following suggestions could improve the quality of the ms.

Comments

  1. It is not clear why polycaprolactone (enriched with 5% hydroxyapatite) membrane should be investigated instead of commercial porcine collagen membrane.
  2. There is no in vitro result and other results related to the basic properties of the membranes.
  3. It is not clear that the PCL group is better than control group in histological results, which seems to be half of the result in this ms.
  4. In the abstract, the short name of guided bone regeneration is GBR, and in the abstract, the short name of guided bone regeneration becomes ROG. ‘80 KDa Molar Mass’ should be written as ‘80 kDa molar mass’ (line 88).
  5. line 284-289, this claim is not true. Please read Materials 2019, 12, 2643.
  6. There are too many format issues, which happens even from the title). For example, Title: ‘Bone Regeneration assessment of Poly-Caprolactone membrane on critical-size defect of rats calvaria.’ If the authors would like to capitalize the title, please keep the rule all the time. ‘Poly-Caprolactone’ should be changed to ‘Polycaprolactone’. The period at the end should be deleted.
  7. Reference. Please put the reference number before the punctuation. And the references are out of date.

Author Response

Dear Reviewer, thank you for all your comments and considerations. We carefully answer all doubts and accepted suggestions.

The answers are attached. 

Reviewer 2 Report

Dear colleagues,

many thanks for submission of the interesting manuscript. Although I think this manuscript is suitable for publication minor revisions have to be conducted.

(1) Please include some information about the (molecular) expectations we nowadays have for GBR membranes such as transmembraneous vascularization etc.

(2) Please describe the histological workup more detailed.

(3) I did not understand the details of your histomorphometrical analysis. Please explain how the inflammatory tissue responses were measured. Did you measure all different cell types? Is it possible using only H&E-stainings?

Many thanks and best wishes

Author Response

Dear reviewer, thank you for all your comments. We carefully answered all doubts and accepted all suggestions.

The answers are attached.

Reviewer 3 Report

Review of Manuscript ID: membranes-1075892 entitled ‘Bone Regeneration assessment of Poly-Caprolactone membrane on critical-size defect of rats calvaria’ by Ana Paula Farnezi Bassi and colleagues.

The above manuscript by Ana Paula Farnezi Bassi and colleagues is a research article on bone regeneration and aimed at aimed to comparatively evaluate the biological performance during bone healing process of 8-mm critical defect on rat’s calvaria using different membranes to assess their influence on the quality of the newly formed bone. In this study seventy-two adult male rats were classified into three experimental groups (n = 24), according to the membranes: the CG– membrane-free control group (only blood clot, negative control), BG – porcine collagen membrane group (Bio-Guide®, positive control), and PCL – poly-caprolactone (enriched with 5% hydroxyapatite) membrane group (experimental group) with histological and histometric analysis done after the treatments. Overall, the authors observed that the PCL membrane was more biocompatible with a potential to help during bone guided regeneration (GBR) procedures.

This manuscript requires major revisions before publication.

Specific Points

  1. The manuscript requires major English language editing. A native English-speaking person must edit this manuscript. There are many instances of spelling mistakes as well as grammar mistakes.
  2. Guided bone regeneration must be written as GBR throughout the manuscript. See several published manuscripts including PMID- 28833567
  3. There are large sections of the manuscript with little or no References at all.
  4. Describe ARRIVE Guidelines please
  5. Animal housing and treatment must be described before euthanization etc. Lines 93-100
  6. Dosages used in study require justification or References. Why choose these dosages.
  7. To emphasize the need for English editing, all animals used in the study are referred to as ‘everyone’ lines 168-169.
  8. Results section- All titles must have descriptive titles. Titles must describe the results obtained. Not just ‘Histological Analysis’ or ‘Inflammatory Response’.
  9. Both Introduction and Discussion must be a minimum of 1.5 pages with more References added.
  10. References used in manuscript must be a minimum of 70.

Author Response

(The authors gave the same response as above.)

Round 2

Reviewer 1 Report

I recommend it for publication based on the revision.

Reviewer 3 Report

Authors addressed all my concerns.